# Re-Use of Caco-2 Monolayers in Permeability Assays—Validation Regarding Cell Monolayer Integrity

**DOI:** 10.3390/pharmaceutics13101563

**Published:** 2021-09-26

**Authors:** Cristiana L. Pires, Catarina Praça, Patrícia A. T. Martins, Ana L. M. Batista de Carvalho, Lino Ferreira, Maria Paula M. Marques, Maria João Moreno

**Affiliations:** 1Coimbra Chemistry Center, Department of Chemistry, University of Coimbra, 3004-535 Coimbra, Portugal; cristiana.lages.pires@gmail.com (C.L.P.); patriciatelesmartins@gmail.com (P.A.T.M.); 2CNC—Centro de Neurociências e Biologia Celular, CIBB—Centro de Inovação em Biomedicina e Biotecnologia, Universidade de Coimbra, 3004-504 Coimbra, Portugal; catarina.op.almeida@gmail.com (C.P.); lino@uc-biotech.pt (L.F.); 3Faculdade de Medicina, Universidade de Coimbra, 3000-370 Coimbra, Portugal; 4Molecular Physical-Chemistry R&D Unit, Department of Chemistry, University of Coimbra, 3004-535 Coimbra, Portugal; almbc@uc.pt (A.L.M.B.d.C.); PMC@ci.uc.pt (M.P.M.M.); 5Department of Life Sciences, University of Coimbra, 3000-456 Coimbra, Portugal

**Keywords:** Caco-2 cells, intestinal permeability, tight junctions, lucifer yellow, high-throughput

## Abstract

Caco-2 monolayers are a common in vitro model used to evaluate human intestinal absorption. The reference protocol requires 21 days post-seeding to establish a stable and confluent cell monolayer, which is used in a single permeability assay during the period of monolayer stability (up to day 30). In this work, we characterize variations in the tightness of the cell monolayer over the stable time interval and evaluate the conditions required for their re-use in permeability assays. The monolayer integrity was assessed through TEER measurements and permeability of the paracellular marker Lucifer Yellow (LY), complemented with nuclei and ZO-1 staining for morphological studies and the presence of tight junctions. Over 150 permeability assays were performed, which showed that manipulation of the cell monolayer in the permeability assay may contribute significantly to the flux of LY, leading to *P*_app_ values that are dependent on the sampling duration. The assay also leads to a small decrease in the cell monolayer TEER, which is fully recovered when cell monolayers are incubated with culture media for two full days. When this procedure is followed, the cell monolayers may be used for permeability assays on days 22, 25, and 28, triplicating the throughput of this important assay.

## 1. Introduction

In vitro cell-based assays are the screening tool of choice for permeability assessment by the pharmaceutical industry, due to the right balance between predictability and ethical issues [1,2]. The Caco-2 cell line, derived from human colonic adenocarcinoma cancer cells, is the most extensively characterized in vitro model for predicting in vivo human oral absorption of novel drugs [3,4,5,6]. Numerous qualitative relationships can be found in the literature regarding the in vitro apparent permeability coefficient (*P*_app_) through Caco-2 monolayers and human jejunal absorption [7,8,9]. Caco-2 cells are not, however, intrinsically intestinal epithelial cells, and this in vitro model shows limitations regarding the metabolizing enzymes and the active transporters expressed [1,10]. When using cell monolayer permeability assays in drug screening, it may therefore be convenient to include more advanced models, such as those derived from intestinal epithelial stem cells [11]. Caco-2 permeability assays are nevertheless excellent in vitro models to predict permeability through passive routes and give important insight regarding active transport [7,8,9,10].

There are several reported protocols for the use of Caco-2 cell lines in permeability assays. They vary in the cell source, passage number, days after seeding in the Transwell™ filter, size and characteristics of the filter, and transport media, among other things. These distinct procedures lead to differences in the properties of the cell monolayer and result in permeability coefficients for a given drug that vary by orders of magnitude [12,13,14]. The variety of procedures is an intrinsic characteristic of scientific research, which is always in search of better and faster protocols. However, this variability limits the use of the extensive amount of literature data for the establishment of quantitative structure/permeability relations. In an attempt to standardize the procedures for well-established assays, the journal Nature Protocols publishes detailed protocols that may be easily followed by the scientific community. This is the case for the protocol defined by Artursson et al. in 2007 regarding the use of Caco-2 monolayers in permeability assays [15]. The major limitation of this reference protocol is its very low throughput due to the long culture period (at least 21 days), although this has been shown to be required to obtain a fully differentiated and confluent monolayer [16,17]. Additionally, in the standard protocol the assay is performed on 12 wells *per* plate, leading to very high implementation costs for the assay of only a small number of compounds [1,15]. In an attempt to accomplish a higher throughput, several groups have developed new protocols, which include accelerated processes for cell differentiation, miniaturization procedures, or the use of several test compounds in each assay.

Filter coating and the use of supplemented culture media has been shown to accelerate cell differentiation, and the post-seeding time has been reduced to 3–7 days [18,19,20,21]. Despite the significant advantages of these shorter-period assay models, several issues have been raised regarding the presence of active transporters and the integrity of the cell monolayer [18,19]. Additional issues with patents and poor clarification of the supplement’s effect on the cell properties have also been pointed out as reasons for the limited use of these accelerated models [21]. Miniaturization to 96- and 384-well plate formats have been employed, allowing for the testing of more compounds and the use of smaller amounts of cells and test drugs [22,23,24]. However, the small cell surface area and the high total surface *per* volume may lead to artifacts due to compound loss by adsorption to the container’s material [25]. Several authors have explored the use of up to ten test compounds in a single permeability assay, based on the assumption that the multiple analytes can be quantified simultaneously with analytical techniques [26,27,28]. Although this approach has been shown to be valid for some groups of test molecules, it cannot be followed for poorly defined compounds due to possible interferences such as, for example, the coexistence of substrates and inhibitors of P-glycoprotein. An interesting compromise is the combination of a well-defined control marker and the unknown test compound, as has been done previously [29,30].

Despite all efforts, only moderate success has been achieved with these higher throughput protocols, and Caco-2 cells grown for 21 days in 12-well plates is still the standard procedure. An analysis of the literature in the last 5 years shows that in 74% of the publications the permeability assay is performed at least 21 days after seeding (only 4% follows accelerated protocols, with ≤7 days after seeding), and 12-well plates are the most common setup. Unfortunately, a large variability is observed for additional parameters such as cell passage number, which has been shown to significantly influence the measured permeability [12,31]. This literature survey shows the widespread use of this permeability assay, with 80% of the published studies on the evaluation of intestinal absorption using Caco-2 monolayers. In addition to its well-established application in pharmacology, this methodology has currently been extended to a broad range of scientific areas. Of particular relevance are the studies related to human nutrition, which account for over 30% of the publications. The wider use of this permeability assay prompts the development of a high throughput methodology, as well as the need for standardization [32].

In the present work, we propose an alternative strategy to increase the throughput of this permeability assay, while following the procedure proposed in the reference protocol [15]. Several studies have shown that the Caco-2 cells maintain their morphofunctional properties from day 21 untill day 30 after seeding [16,17,33]. The maintenance of the cell monolayer properties for this extended period has been used to add flexibility in the choice of the day of the experiment while performing a single permeability assay. Although it has never been explored, this stability also points towards the possibility of performing several consecutive assays. The permeability assay requires manipulation of the cell monolayer and exposure to the tested compounds and transport media, which may lead to perturbations to the monolayer integrity. The use of non-toxic concentrations of the tested compounds is a general requirement in any permeability assay, and it can be easily pre-evaluated using standard assays, such as MTT [34]. The major concern regarding the re-use of a cell monolayer in consecutive permeability assays is therefore the possible loss of cell monolayer integrity due to continued manipulation of the filter. Development of methodologies for the re-use of the cell monolayers must first guarantee a full recovery of the cell monolayer properties between assays. 

The aim of this study is three-fold: (i) to evaluate the effect of the permeability assay on the monolayer integrity; (ii) to evaluate whether cell incubation with culture media is sufficient for recovery of the monolayer integrity; (iii) to determine the optimal duration of the interval between consecutive permeability assays. Assessment of the monolayer integrity was based on its transepithelial electrical resistance (TEER), on the permeability of the well-known marker of the paracellular pathway Lucifer Yellow (LY), and on the distribution of the tight junction protein zonula occludens-1 (ZO-1). A preliminary evaluation showed that one day of incubation with culture media is not enough for a full recovery. An extensive characterization has therefore been performed for a two-day incubation period, with consecutive permeability assays being performed at days 22, 25, and 28 after seeding. Single sampling (at 60 min) and multi-time sampling (10, 20, 30, and 60 min) was evaluated to support the applicability of the cell monolayer re-use when following both procedures. 

More than 150 permeability assays were performed, and over 50 cell monolayers were characterized by confocal microscopy. With this extensive work, the statistical distribution of the permeability parameters could be described, allowing a quantitative comparison between the distinct procedures and conditions. The results obtained show that the re-used cell monolayers maintain their integrity when following the proposed protocol, triplicating the throughput of previously reported permeability assays using Caco-2 monolayers.

## 2. Materials and Methods

### 2.1. Reagents and Materials

Caco-2 cells were obtained from the European Collection of Authenticated Cell Cultures (ECACC 09042001, Salisbury, UK). Dulbecco’s modified Eagle’s medium (DMEM) (4.5 g/L d-glucose, l-glutamine, without pyruvate), nonessential amino acids (NEAA), penicillin 10,000 U mL^−1^/streptomycin 10,000 µg mL^−1^ solution (Pen/Strep), 0.25% (*w/v*) Trypsin, ethylenediamine tetraacetic acid (EDTA, ≥98.5%) Triton X-100, sodium bicarbonate (≥99.7%), Hank’s balanced salt solution (HBSS), 4-2-hydroxyethyl-1-piperazineethanesulfonic acid (HEPES), and Lucifer yellow CH di-potassium salt (≥99%) were purchased from Sigma-Aldrich Química S.A. (Sintra, Portugal). Fetal bovine serum (FBS) was obtained from Gibco-Life Technologies (Porto, Portugal). Corning^®^ Transwell™ 12-well inserts with a polycarbonate membrane (1.12 cm^2^ surface area, 0.4 µm pore size) and 12-well cell culture plates were obtained from VWR (Lisboa, Portugal). The primary and secondary antibodies were purchased from Alfagene (Lisboa, Portugal), Dako fluorescence mounting medium from Agilent (Lisboa, Portugal), and bovine serum albumin (BSA) from Applichem (Darmstadt, Germany). Ammonium formate (≥99%) and organic solvents of analytical grade were acquired from Fischer Scientific (Lisboa, Portugal).

### 2.2. Caco-2 Cell Culture and Seeding

Caco-2 cells culture and monolayers preparation was as described in the reference protocol [15]. Cells were maintained in T75 cm^2^ flasks in culture medium (DMEM supplemented with 1% (*v/v*) NEAA, 10% (*v/v*) heat-inactivated FBS), in a humidified atmosphere of 5% CO_2_ at 37 °C. As recommended in the reference protocol, the permeability assays were performed using cells in passages 95–105. The cells were thawed on passage 89 and sub-cultured twice a week at or shortly before 90% confluency by a 1:8 split. To avoid seeding aggregated cells and the formation of multilayers in Transwell™ inserts, the cell clusters were disaggregated into single cells by passing them 3 times through a syringe needle of 23 gauges (BD Falcon) before cell counting. Caco-2 cells were seeded at 2.6 × 10^5^ cells/cm^2^ on 12-well polycarbonate filter inserts. The culture medium (with 1% (*v/v*) Pen/Strep) was replaced by fresh medium 6 h following seeding and then every 2–3 days thereafter until transport experiments.

### 2.3. Permeability Assays and Transepithelial Electrical Resistance (TEER)

The reference protocol [15] was followed in the permeability assays, with slight modifications as required by the new experimental approach. Manipulation of the cell monolayers was conducted under sterile conditions (in the laminar flow hood or incubator). At the day of the experiments, the culture medium was removed and HBSS (further supplemented with 25 mM HEPES, 0.35 g/L sodium bicarbonate, pH 7.4) pre-warmed to 37 °C was added to the donor (0.5 mL) and receiver (1.5 mL) sides to wash the cell monolayers. The plate containing the inserts was placed (lid-covered) in an incubator without CO_2_ at 37 °C with an orbital shaker (IKA-Schüttler MTS_4_, JMGS, Lisboa, Portugal) at 50 rpm for 8 min. The washing step was repeated with new HBSS. The TEER was measured in HBSS at 37 °C using a Millicell^®^ ERS-2 voltmeter equipped with a chopstick electrode pair (Merck, Lisboa, Portugal). A filter insert without cells was included in each set of experiments for correction of the TEER value for the background resistance. The resistance values (Ω cm^2^) of the cell monolayers were obtained by subtracting the TEER value from blank inserts and multiplying by the surface area of the insert. All cell monolayers presented TEER values above 200 Ω cm^2^. At least one insert of the set was fixed and stained for confocal microscopy analysis (see next section). The remaining ones were used to perform the apical to basolateral permeability assays with the paracellular marker LY. For this, the donor HBSS washing solution was decanted, and the inserts were placed into empty wells of a fresh 12-well plate. The permeability assay was initiated with the addition of 450 µL of the LY solution (20 µM in HBSS at 37 °C) simultaneously to the donor compartments of 3 to 4 insert, from which a 50 µL aliquot was immediately collected and stored for the calculation of the solute concentration in the donor compartment at t = 0. The inserts containing the LY solution were subsequently positioned in a pre-prepared 12-well plate containing HBSS (1.2 mL *per* well), and the lid-covered plate was placed on the incubator (without CO_2_, 50 rpm and 37 °C). At the defined sampling times, the plate was positioned into the laminar flow hood and the inserts were transferred into new wells containing fresh HBSS. The high fluorescence quantum yield of LY allows its detection by HPLC, even for the very small amount expected to permeate during short permeation intervals. In this work we used 10, 20, 30, and 60 min or 60 min only as sampling times. At the end of the assay, inserts were transferred to empty wells and 50 µL was taken from the donor compartment for the calculation of the solute mass balance. The remaining solution was decanted, the inserts were placed on new wells containing 1.5 mL of pre-warm HBSS, and 0.5 mL of pre-warmed HBSS was added to each insert for TEER measurements. Some selected monolayers were prepared for microscopy; the remaining were incubated with culture media for re-use, as described below. 

Incubation of the cell monolayer with HBSS and the manipulation required for the permeability assays may lead to perturbation of the monolayer integrity. In fact, we (and others [29,35]) have observed a small but systematic decrease in the TEER value after the permeability assay. Hence, for the re-use of the cell monolayer in additional permeability assays, it is necessary to first ensure that the monolayer integrity is fully recovered. One hypothesis of this work is that this may be achieved by incubation of the cell monolayer in the culture media. Preliminary experiments showed that one day of incubation was not enough to re-establish the cell monolayer integrity (results not shown). In the work currently described, the cells were incubated for 2 days in supplemented DMEM in a humidified atmosphere of 5% CO_2_ at 37 °C. To maximize the use of the already established cell monolayers up to three times during the total period where the monolayers are stable (21–30 days) [16,17,33], the days selected for the permeability experiments were 22, 25, and 28 post-seeding. 

The apparent permeability (*P*_app_) of LY was determined from the amount of solute transported across the monolayer per time:(1)$$P_{app}=\frac{\frac{\Delta Q^{A}}{\Delta t}V^{D}}{AQ_{0}^{D}}$$
where $\Delta Q^{A}$ is the amount of solute that enters the acceptor compartment during the time interval, Δ*t* (in seconds), A is the surface area of the filter (1.12 cm^2^), $V^{D}$ is the volume of the donor compartment (0.4 cm^3^), and $Q_{0}^{D}$ is the amount of solute in the donor compartment at the beginning of the time interval considered. When several sampling times were performed, the amount of solute in the donor compartment at the beginning of the time interval considered was calculated from that in the beginning of the experiment by subtracting the solute that reached the acceptor compartment in the preceding sampling times. This equation is equivalent to that indicated in the reference protocol for the instantaneous permeability coefficient [15]. The only difference is that, instead of directly considering the initial concentration of the tested compound in the donor compartment, we explicitly indicate the amount of compound and the volume of the compartment. This avoids subjectivity when using different units for the distinct variables. In Equation (1), the amount of solute may be in the most convenient units, provided that the same units are used for the solute in both compartments ($\Delta Q^{A}$ and $Q_{0}^{D}$). Additionally, it becomes more intuitive that the surface and the volume must use the same length units, which will be that of the resulting *P*_app_.

### 2.4. LY Quantification

At the conclusion of the permeability experiment, samples were stored at −20 °C until analysis by HPLC. LY quantification was performed by reversed-phase HPLC (Agilent 1200), consisting of a quaternary solvent pump, a fluorescence detector (model G1321A), an auto-sampler with a 900 µL injection loop, and a Zorbax C18 column (250 × 4.6 mm, 5 µm particle) at 23 °C. The mobile phase for LY consisted of 100% of ammonium formate 50 mM (pH 3.5 adjusted with formic acid) at a flow rate of 1 mL/ min. The injection volume was 100 μL for the donor (previously diluted 1:20 with 950 μL of HBSS) and 900 μL for the samples from the acceptor compartment. The fluorescence detector was set to λ ex = 430 nm and λ em = 530 nm. The LY retention time was 7.9 min, and the total analysis time per sample was 10 min. Independent calibration curves were performed for 100 μL and 900 μL injections, and a linear dependence (r^2^ ≥ 0.9998) was obtained in the concentration range from 0.001 to 1 µM and 0.001 to 0.2 µM, respectively (see Appendix A for details). The lower limit of quantification was 0.02 µM for 100 μL injections and 0.0005 µM for 900 μL. The corresponding limits of detection were 0.006 µM and 0.0002 µM.

### 2.5. Confocal Laser Screening Microscopy

For morphological studies by confocal laser screening microscopy (CLSM), Caco-2 monolayers were stained directly on the Transwell™ inserts, before or after the permeability assays. Cell monolayers were first fixed with 4% (*w/v*) paraformaldehyde in PBS for 10 min. After washing three times with PBS (incubated for 5 min), cells were permeabilized for 10 min using 0.1% (*w*/*v*) Triton X-100 in PBS and then washed again. Nonspecific binding sites in the samples were subsequently blocked with 1% (*w*/*v*) BSA in PBS for 30 min. The tight junction protein ZO-1 was stained with ZO-1 rabbit polyclonal antibody at a dilution of 1:200 overnight at 4 °C. After incubation, cells were washed 3 times with PBS and incubated with a secondary antibody, cyanine 3 conjugated goat anti-rabbit IgG, at a dilution of 1:100 for 30 min at room temperature, protected from light. Cell nucleus was stained with Hoechst 33342 dye, 1 μg/mL, for 10 min. All stains were prepared in PBS containing 1% (*w*/*v*) of BSA. At the end of this incubation, the cells were washed with PBS and the filter was cut from the insert and mounted cell side up on a glass slide using Dako fluorescence mounting medium.

Confocal images of the slides were captured on a Zeiss LSM 710 (Carl Zeiss MicroImaging, GmbH) inverted confocal microscope at 20× magnification. A Z-stack was performed with a z-interval of 1 μm, the cell layer being typically enclosed in 8 or 9 optical slices indicating the formation of a single cell layer [36]. Image acquisition and analysis of maximal intensity Z projection of these stacks was performed using Zen 2012 software (Black edition, version 1.1.2.0, Zeiss). 

A total of 52 confocal images of Caco-2 monolayers were characterized, covering several conditions including the culture day, effects of the permeability assay, and single use/re-use of the cell monolayer. ImageJ software (version 1.52, NIH, USA) was used to determine the cell density and the fractional area occupied by nuclei and by the tight junction protein ZO-1. The image area (428 × 428 µm^2^) was analyzed on at least 2 independent cell monolayers *per* condition. Each image was manually threshold to select nuclei and ZO-1 only fluorescence.

### 2.6. Statistical Analysis

Experiments were carried out at least in triplicate (wells per plate) and were independently repeated at least two times in different cell batches and cell passages. The very large number of assays performed allowed for the characterization of the parameter statistical distribution, which was shown to be a LogNormal for the permeability coefficient. The characteristic values and uncertainty associated with this parameter cannot therefore be obtained directly from the average and standard deviation of the values measured. Instead, the most probable value was obtained from the average of Log*P*_app_, and uncertainty is expressed as 95% confidence intervals, calculated from the standard deviation of Log*P*_app_ [37,38]. 

A multivariate analysis (MVA) was applied to assess the correlation between the parameters and all potentially relevant independent variables. The multivariate regression model used to fit the data is given below: (2)Y=β0+β1×X1+β2×X2+…+βi×Xi
where the dependent variable (*Y*) is described as a linear function of the independent variables (*X*i). *β*0 is the *Y*-intercept, and each coefficient (*β*i) reflects the effect of the corresponding independent variable (*X*i) on *Y*. The values of *β*i are estimate by minimizing the sum-of-squares of the differences between the values of *Y* predicted by the equation and the *Y* values in the data. The coefficients have the units of *Y* divided by the units of the corresponding *X*. The confidence interval CI_95_ and the *p*-value were calculated for each *β*i; *p* < 0.05 was considered as statistically different from zero. This statistical analysis was performed using GraphPad Prism (version 8.4.2, San Diego, CA, USA).

## 3. Results and Discussion

### 3.1. Effect of Re-Use on the Cell Monolayer Integrity after Single Time Sampling Permeability Assays

Caco-2 monolayers have previously been shown to be stable and adequate for performing transport studies from day 21 to day 30 after seeding in the Transwell™ filter [16,17,33]. In this work, we evaluate whether the same cell monolayer may be re-used in consecutive permeability assays during this interval. Preliminary results showed that one day between the assays was not enough for full recovery of cell monolayer integrity. Therefore, the systematic study presented in this work was performed with an interval of two full days between consecutive permeability assays. The selected days for the assays were 22, 25, and 28. 

To distinguish between the effects of performing the permeability assay from eventual variations of the cell monolayer properties between day 22 and 28, single use assays were also performed on day 25 and day 28. The tightness of the cell monolayer was first evaluated by measuring its TEER, both before and after the permeability assays. The very large number of assays performed in this work permit the characterization of the distribution probability of TEER values, and the results obtained before the execution of the permeability assay are shown in Figure 1A. Broad distributions are obtained at all days, with a small increase at day 28 (best fit Normal distribution with µ ± σ = 1228 ± 248 Ω cm^2^) when compared to day 22 and day 25, which show a similar value of TEER (1036 ± 491 and 986 ± 432 Ω cm^2^, respectively), the overall distribution at all days being 1067 ± 473 Ω cm^2^. The distribution of TEER values after the execution of a permeability assay are represented in Figure 1B. It is observed that there is a small decrease in the mean value of TEER (766 ± 383 Ω cm^2^, for the best fit distribution of the cumulative data at all days), which is mostly due to a decrease in the TEER values obtained after the permeability experiment for cells on day 22 (from 1036 to 719 Ω cm^2^), but also for cells on day 28. Nevertheless, TEER values above the threshold of 200 Ω cm^2^, which is commonly accepted for a confluent cell monolayer [39,40], are obtained in all cases; see Appendix A for further details.

After the execution of the permeability assay, the cell monolayers were incubated in culture media for two full days to evaluate if their barrier properties could be re-established. The results obtained for TEER are shown in Figure 2. The statistical distributions obtained for the TEER values of cell monolayers not previously used on permeability assays are undistinguishable from those of the cell monolayers previously used and incubated in culture media for two full days. This indicates that the 2 days of incubation with the culture medium allows for the re-establishment of the cell monolayer integrity, overcoming the small perturbations induced by the permeability assay (including manipulation and incubation in HBSS). 

The analysis above is based on the average values obtained for each experimental condition. A closer look at the results for each individual cell monolayer is presented in the Appendix A and shows that: (i) a large decrease in the value of TEER after the permeability assay was only observed in 1/3 of the cell monolayers, while in the vast majority of the inserts, TEER varied by less than 20% (increase or decrease); (ii) from day 22 to 25 (before the permeability assay in both days), the TEER value increased for all inserts. Thus, the decrease observed after the assay on day 22 was fully recovered for all inserts. When analyzing the results between the 2nd and the 3rd use of the cell monolayer in the permeability assays, a full recovery of TEER was also observed in the majority of the inserts. 

The results obtained with TEER suggest that the procedure developed in this work leads to full recovery of the cell monolayer integrity and that they can be re-used in permeability assays at days 25 and/or 28 after being used on day 22. However, due to the high variability in this parameter, it may be questioned if it accurately reflects the integrity of the cell monolayer [16,36,41,42]. We have therefore also evaluated the effect of monolayer re-use through the results obtained for the permeability of a paracellular marker. The results obtained for LY *P*_app_, calculated from a single 60 min sampling interval, are shown in Figure 3.

At day 22, a large number of permeability assays were performed (N = 34), leading to a well-defined distribution frequency. A clear misfit is observed when the best fit of a Normal distribution is performed (grey dashed line), with the experimental data showing strong asymmetry towards high *P*_app_ values. The quality of the best fit is significantly improved when a LogNormal distribution is considered (black line), which is in fact the statistical distribution expected for rate constants [37]. Due to the asymmetry in the distribution, the uncertainty associated with this parameter should be expressed as a confidence interval [37]. This interval may be calculated from the parameters of the LogNormal distribution that best describes the results, or directly from the average and standard deviation of the observed values of Log*P*_app_, leading, respectively, to [1.1, 4.0] × 10^−7^ cm/s and [1.2, 3.9] × 10^−7^ cm/s at 95% confidence (CI_95_), and an average value of *P*_app_ equal to 2.1 × 10^−7^ cm/s for both analyses. 

The distribution frequency of the *P*_app_ values obtained with cell monolayers on days 25 and 28 is also shown in Figure 3. The number of assays performed with cell monolayers on day 25 that have been previously used on day 22 is large (N = 23) and leads to a well-defined distribution frequency (light green). However, the number of assays performed on day 25 for cell monolayers used on a single day (dark green, N = 6) is too small to allow the characterization of the frequency distribution. The characteristic value and confidence intervals must therefore be calculated from the average and standard deviation of the observed values of Log *P*_app_, leading to µ = 1.8 × 10^−7^ cm/s and CI_95_ equal to [0.6, 5.3] × 10^−7^ cm/s. The same average value is also obtained for *P*_app_ on re-used cell monolayers. In this case, the much larger number of assays leads to a smaller uncertainty, with the CI_95_ being [1.1, 2.6] × 10^−7^ cm/s. This shows that the two conditions do not lead to statistically different results and, thus, the cell monolayers that have been used on day 22 may be re-used on day 25 if following the proposed protocol. When this treatment is performed for cells on day 28 post-seeding, the parameters obtained are µ = 3.2 × 10^−7^ cm/s and CI_95_ equal to [1.7, 6.1] × 10^−7^ cm/s for cell monolayers used a single time (N = 6) and µ = 1.6 × 10^−7^ cm/s and CI_95_ equal to [0.7, 3.6] × 10^−7^ cm/s for cell monolayers previously used on day 22 and 25 (N = 18). As observed on day 25, the small number of assays performed with single use cell monolayers lead to a large uncertainty, and the two conditions are not statistically different. The collective result led to µ = 1.9 × 10^−7^ cm/s and CI_95_ equal to [0.8, 4.7] × 10^−7^ cm/s. When the results obtained on all days are analyzed (right plot in Figure 3), it is observed that they are very well described by a single LogNormal distribution. This shows that, although a small decrease is observed in the *P*_app_ values as cell monolayers are maintained on the inserts (specially at day 28), the distributions are not statistically independent. The parameters obtained from the collective values at all days and conditions are µ = 2.0 × 10^−7^ cm/s and CI_95_ equal to [1.0, 3.8] × 10^−7^ cm/s. The small values obtained for LY permeability show that the cell monolayers are intact and tightly sealed at all conditions, *P*_app_ < 5 × 10^−7^ cm/s [15,39,43]. Thus, the results show that, when the cell monolayer is allowed to recover in culture media for 2 full days between assays, its integrity is not compromised, and they may be re-used on days 25 and 28. 

### 3.2. Effect of Multi-Time Sampling on the Paracellular Permeability through Caco-2 Monolayers

For the evaluation of new compounds’ permeability, the use of several time points is usually required in order to allow the characterization of fast permeating compounds, while sink conditions are guaranteed. The sampling may be performed either with the replacement method (removing part of the solution in the acceptor compartment and replacing it with fresh transport media), or with the transfer method (where the insert is transferred to new wells containing fresh transport media). The latter is preferable for fast permeating solutes, as it takes only a few seconds to transfer several inserts and no test compound is present in the acceptor compartment at the beginning of each new time interval. It is also essential to assess the monolayer integrity, which is often achieved using a control solute such as LY, which permeates paracellularly [32,40,44,45]. This test may be carried out in dedicated cell monolayer inserts prepared from the same batch of cells, or in the same monolayer insert, before or after the test compound (in this case providing additional evidence for eventual toxicity) [46]. With a view to reducing the duration of the incubations with the transport media and the manipulations of the cell monolayers, a better approach is to evaluate the permeability of both the control (LY) and test compounds in the same Transwell™ [30,34]. This procedure has the advantage of including a quality control for each filter, thus facilitating the identification of cell monolayers with compromised integrity (outliers), and therefore decreasing the variability. In this approach, it is necessary to ensure that the compounds do not interfere with each other during their permeation through the cell monolayer. LY is anticipated to be a good control compound in this respect because its high polarity prevents association with the cell membrane and will therefore avoid interference with membrane permeating solutes [29]. Additionally, it is not a substrate of transporters in the membrane (influx or efflux) [47]. 

In this section we evaluate the impact of multi-time sampling and different time intervals in the permeability of LY. The inserts were transferred to wells containing pre-warmed fresh HBSS at time points 10, 20, 30, and 60 min. The results obtained for the instantaneous permeability (Equation (1)) during the first 10 min sampling interval are shown in Figure 4 (upper plots), together with those previously obtained for the single time point sampling at 60 min. As observed before (Figure 3), the distribution frequency obtained for a 10 min sampling time interval is best described by a LogNormal distribution (a Normal distribution for the parameter represented in Figure 4, Log*P*_app_). Independent distributions, however, are observed for both sampling time intervals, with higher *P*_app_ values obtained for the shorter sampling time interval (µ = 15 × 10^−7^ cm/s and CI_95_ equal to [7.6, 29] × 10^−7^ cm/s). 

An alternative procedure to quantify permeability is directly through the amount of LY that reaches the acceptor compartment (*Q*_A_) during the sample time interval (Figure 4, lower plots). In this case, it is observed that a Normal distribution centered at around −0.6 is obtained for Log*Q*_A_ during both sampling intervals (10 or 60 min), corresponding to *Q*_A_ around 0.2% (for details see the Appendix A). A very similar amount of LY was also observed to permeate through Caco-2 monolayers on day 23 post-seeding during a 90 min sampling interval, 0.27% [30]. This shows that, for cell monolayers on day 22 post-seeding, manipulation of the cell monolayer during the permeability assay leads to a significant amount of LY transport and that very little permeation is observed during the sampling time interval, thus leading to *P*_app_ values that are strongly dependent on the sampling time interval. When assessing the integrity of the cell monolayer using distinct incubation times, it is therefore preferable to refer to the observed permeability in terms of % of the control solute transported, which is usually indicated as below 0.5% for a tight Caco-2 monolayer [48].

Very similar results are observed for cell monolayers on day 25 post-seeding (Figure 4, middle plots). When considering cell monolayers on day 28 (right plots), the values obtained for LY *P*_app_ are still dependent on the sampling time interval, but a significant overlap is observed between the frequency distribution obtained for both conditions. Conversely, the frequency distributions of the amount of LY that permeates shows significant differences between the two sampling time intervals, being higher for the longer sampling interval. 

A closer look on the effect of days post-seeding in the amount of LY that permeates through cell monolayers shows that the major variation is observed for the 10 min time sampling, with a decrease from µ = 0.25% at day 22 to 0.19% at day 25, and 0.09% at day 28. This indicates that the contribution from the manipulation of the insert is very substantial on day 22 and becomes less significant for cell monolayers maintained in the insert for longer times, supporting an increase in the cell monolayer tightness, as previously suggested by the small increase in TEER (Figure 1 and Appendix A). Therefore, when comparing the values of *P*_app_ for different analytes, it is important that the same conditions are used in the permeability assays, namely the day post-seeding and, most importantly, the sampling time intervals.

We have also analyzed the effect of multi-time sampling on the observed instantaneous amount of LY that permeates and the corresponding value of *P*_app_; the results obtained for the amount of LY that reaches the acceptor compartment are shown in Figure 5. A small increase in *Q*_A_ is observed for consecutive sampling with the same time interval: µ = 0.25, 0.29 and µ = 0.37% on day 22; µ = 0.19, 0.25 and µ = 0.25% on day 25; and µ = 0.09, 0.10, and µ = 0.13% on day 28. An increase is also observed on the width of confidence intervals (see Appendix A). This shows that repeated manipulation of the cell monolayer leads to more significant perturbation and that the effect is not the same for all cell monolayers. When the cumulative amount of LY that reaches the acceptor compartment during the first 30 min (with sampling at each 10 min) is analyzed, an upward curvature is therefore observed (Appendix A). However, when the final sampling at 60 min (Δt = 30 min) is included in the cumulative transport, the non-proportionality between *Q*_A_ and the sampling time interval dominates, and a downward curvature is observed for the whole 60 min sampling.

The results shown above have been obtained with transfer of the insert into wells containing fresh transport media at the end of each sampling interval. During transfer, the level of transport media across the cell monolayer is not balanced, with a hydrostatic pressure being applied on the cell monolayer in the direction apical-to-basolateral (top-to-bottom compartment). The results presented show that this force may lead to a significant amount of analyte transport, at least for analytes that permeate very slowly and paracellularly, as is the case of LY. To evaluate if this effect was due to the approach followed (transfer), some experiments were performed with sampling evaluated following the replacement approach (where 600 µL are withdrawn from the acceptor compartment at the selected sampling times and replaced by fresh transport media). Equivalent results were obtained. This can be understood because, when using the replacement approach, the cell monolayer is also subject to an unbalanced hydrostatic pressure during the time between the removal of transport media from the acceptor compartment and its replacement. The duration of this unbalance is in fact longer than when following the transfer approach. Additionally, the analysis of data from the literature shows that small deviations from linearity are usually observed in the cumulative amount of analyte transported (both when using the transfer and the replacement approach) [49,50,51,52].

### 3.3. Morphological Features and Integrity of the Cell Monolayer

The effects of the day post-seeding and execution of permeability assays on the morphological features and integrity of the cell monolayer were also characterized by CLSM, both for cell monolayers used on a single assay or re-used. Representative images of the tight junctions at each condition are displayed in Figure 6, and a larger area including the nuclei is shown in the Appendix A. A quantitative analysis is discussed in Section 3.4.

All cell monolayers show an extensive network of tight junctions, as stained by ZO-1, and a similar cell density. This suggests that the execution of the permeability assays does not lead to significant perturbations of the cell monolayers, which is not in agreement with the results obtained with the permeability assays that show a small increase in the leakiness for multi-time sampling. It should be noted, however, that the permeating solutes are several orders of magnitude smaller than the resolution of confocal microscopy (a few Å as compared with several hundreds of nm). Although the tight junction’s network remains apparently intact when observed at a microscopic resolution, its integrity may be compromised, allowing the passage of small molecules. In this respect, it is important to consider the size of the pores in the tight junctions of confluent Caco-2 monolayers. Using a large set of paracellularly permeating solutes of different radii, Adson et al. obtained an effective pore radius of 12 Å for Caco-2 monolayers at days 21–23 after seeding [43]. Similar results have recently been obtained by other authors, with an effective pore radius varying between 6 and 16 Å [53,54]. A strong correlation has also been obtained between the tight junctions’ pore radius and the LY permeability [53]. From this relationship, the *P*_app_ values obtained in the present work suggest a pore radius of approximately 6 Å. Given the several orders of magnitude difference between the resolution of the confocal images, and the tight junctions´ pores through which LY permeates, it is not surprising that a continuous tight junction’s network was observed despite the increased LY permeability.

To provide information regarding the transversal position of the tight junction protein ZO-1, the z-stack projection for the filters with monolayers on day 28 post-seeding is show in Figure 7. It is observed that ZO-1 is located only at the apical side in all conditions. Therefore, the execution of the permeability assays does not lead to a redistribution of tight junctions, even when three permeability assays are performed with the same cell monolayer.

### 3.4. Multivariate Analysis

The very large set of experiments presented in this study (N = 156) was performed over several months, using a large set of cell monolayers (three batches at passages 95 to 105, see Appendix A for LY *P*_app_ obtained for cell monolayers at day 22 post-seeding). This introduces some uncertainty in the results, but significantly improves the robustness of the conclusions achieved. A multivariate analysis has been performed to evaluate the correlation between the permeability parameters and the independent variables. When TEER before the permeability assay is considered, the independent variables are the cell batch, cell passage number (between 95 and 105), day post-seeding, re-use of the cell monolayers, and sampling time points used on the previous permeability assays. For the TEER value after the permeability assay, the variable TEER before the assay was also included as an independent variable, and the TEER after the assay was also considered when the multivariate analysis was performed for LY transport (quantified by *P*_app_ and *Q*_A_). The results obtained are presented in the Appendix A, a brief discussion of the major findings is given here.

The value of TEER before the permeability assay was found to be strongly correlated only with the cell batch, while the correlation of TEER after the assay with that before the assay is the only one that is statistically significant. However, the amount of LY transported showed no significant correlation between the value of TEER (both before and after the permeability assay), indicating that the two variables are reporting distinct properties of the cell monolayer. On the other hand, when LY transport is quantified by its *P*_app_, a strong correlation is observed with the duration of the sampling interval, Δt, smaller sampling intervals leading to higher LY *P*_app_. As discussed in Section 3.2, a small negative correlation (not statistically significant) is observed for *Q*_A_. A significant correlation is also observed between *P*_app_ and the day post-seeding (cell monolayers becoming more impermeable to LY from day 22 to 28).

The nuclei and the tight junction protein ZO-1 of 52 cell monolayers were stained and analyzed by confocal microscopy. The variables considered were the cell density (number of nuclei in an area with 428 × 428 µm^2^), the area occupied by nuclei, and that occupied by tight junctions containing ZO-1. The results obtained are shown in the Appendix A. A high variability is observed for the cell density: 3.1 (±0.7) × 10^5^ cells/cm^2^ at day 22 before any permeability assay (N = 7) and 3.3 (±0.8) × 10^5^ cells/cm^2^ for all cell monolayers (N = 52). The correlation is not, however, statistically significant with cell batch, passage number, day post-seeding, or cell monolayer use/re-use. As expected, a positive correlation is obtained between the cell density and the area occupied by the nuclei. This correlation is, however, sub-linear, which indicates that cells (and their nucleus) are more elongated when at higher density and is consistent with a continuous cell monolayer in all situations. A positive correlation was also observed between the cell density and the cell monolayer TEER, suggesting that the thickness of the monolayer may influence the value of TEER obtained. This correlation was strong when only the cell monolayers at day 22 (N = 7) were considered (See Appendix A) but become not statistically significant when cell monolayers at all conditions were considered (for details see the Appendix A). No correlation, however, is observed between the area occupied by nuclei and by tight junctions, or with the amount of LY that has permeated during the assay. This shows that TEER is affected by parameters not related to the cell monolayer tightness, this being better evaluated by the permeability of paracellular markers such as LY. The area occupied by tight junctions does not show statistically significant correlations (or any weak correlation) with any of the variables considered.

Overall, the results obtained show that the cell monolayer properties are not affected by their re-use at days 25 and 28 when maintained in culture media for two full days between the permeability assays.

## 4. Conclusions

The present study shows that, after using a Caco-2 monolayer in a permeability assay, two days of incubation in culture media are required and are sufficient for the re-establishment of the cell monolayer integrity. This enables the re-use of the cell monolayer in two additional permeability assays, triplicating the throughput of this in vitro model relative to the reference protocol.

The large set of permeability assays performed in this study allowed the characterization of the frequency distribution of the *P*_app_ parameter, which was shown to be skewed towards higher values following a LogNormal distribution. The characteristic value and associated uncertainty cannot therefore be obtained from the average and standard deviation of the observed values of *P*_app_. Instead, the more probable *P*_app_ should be calculated from the average of Log*P*_app_ and the confidence intervals obtained from the standard deviation of Log*P*_app_ and the number of assays [37,38].

It is also observed that manipulation of the cell monolayer during the permeability assay contributes significantly to the amount of analyte transported from the apical (top) to the basolateral (bottom) compartment, at least for solutes that permeate through the paracellular route. This leading to permeability coefficients strongly dependent on the incubation time. It is concluded that the integrity of the cell monolayer should be evaluated through the amount of paracellular marker that reaches the acceptor compartment during the sampling period, and not by the calculated instantaneous permeability coefficient.

Another important conclusion of this work is that the leakiness of the cell monolayers decreases from day 22 to 28 (both for cell monolayers used on a single permeability assay or re-used following the proposed protocol), and that the contribution of cell manipulation to the observed transport is less significant for cell monolayers on day 28 post-seeding. Therefore, cell monolayers on day 28 may be preferable when characterizing the permeability of a set of analytes requiring different sampling time intervals.

The full validation of the methodology proposed for the re-use of the Caco-2 cell monolayer in consecutive permeability assays requires the characterization of the transport of analytes permeating through distinct routes, including passive and transporter mediated influx as well as efflux. The validation regarding cell monolayer integrity performed in this work represents the first step towards this goal. Future contributions, both by ourselves and by the scientific community, will be important in order to perform the extensive work required for the full validation of the protocol, increasing the throughput of Caco-2 permeability assays, while following the reference protocol.

## Figures and Tables

**Figure 1 pharmaceutics-13-01563-f001:**
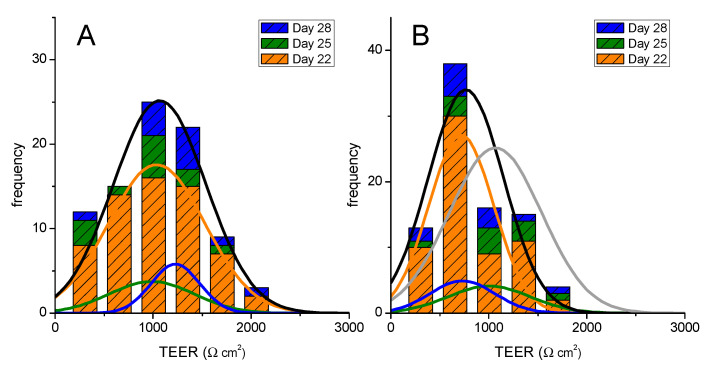
Evaluation of the Caco-2 monolayer integrity when used for a single permeability assay at distinct days after seeding (22, 25, and 28). The TEER values obtained before are shown in Plot (**A**), and after the permeability assay in Plot (**B**). The lines are the best fit of a Normal distribution to the results obtained on each day (colored lines) or cumulatively at all days (black line). The grey line in Plot B is the overall distribution obtained before the permeability experiment.

**Figure 2 pharmaceutics-13-01563-f002:**
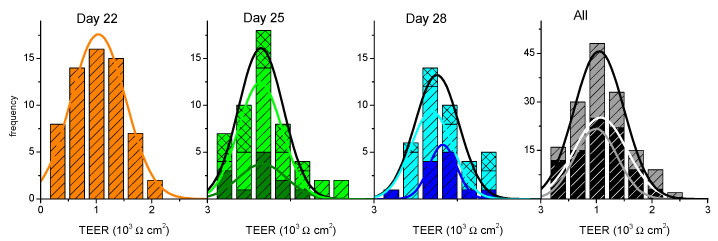
Evaluation of the re-establishment of Caco-2 monolayer integrity when incubated in culture media for two full days after being used for a permeability assay. The TEER values for cell monolayers not used on permeability assays are shown in dark colors (
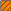
, 
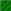
, 
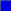
), and those previously used are shown in light colors (
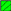
 and 
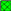
, TEER on day 25 previously used on day 22; 
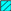
 and 
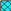
, TEER on day 28 previously used on days 22 and 25, on single 
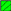


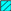
 or multi-time 
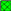


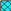
 sampling). The lines are the best fit of a Normal distribution to the results obtained on single use or re-used cell monolayers (colored lines) or cumulatively for each day (black line), with the parameters given in the Appendix A. The cumulative results at all days are shown on the right plot (

, TEER before assay on days 22, 25, or 28; 

, TEER before assay on days 25 or 28 after being used on day 22 or on days 22 and 25).

**Figure 3 pharmaceutics-13-01563-f003:**
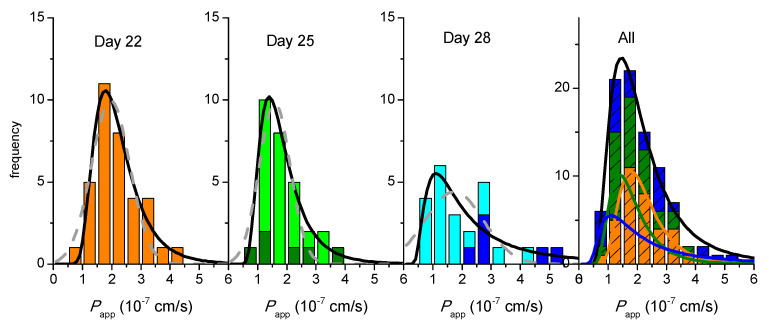
Dependence of LY *P*_app_ (single sampling at 60 min) on the day post-seeding and on the re-use of the monolayer for additional permeability assays after incubation with culture media for two full days. The left plots show the distribution of *P*_app_ values obtained on the different days after cell seeding; dark colors represent results for single use and light colors for re-used cell monolayers; the lines are the best fit of a Normal distribution (grey dashed) or a LogNormal distribution (black continuous). The plot on the right shows the cumulative results at all days and conditions; the lines are the best fit of a LogNormal distribution for data from each day (colored) and collectively for all days (black), with the parameters given in the Appendix A.

**Figure 4 pharmaceutics-13-01563-f004:**
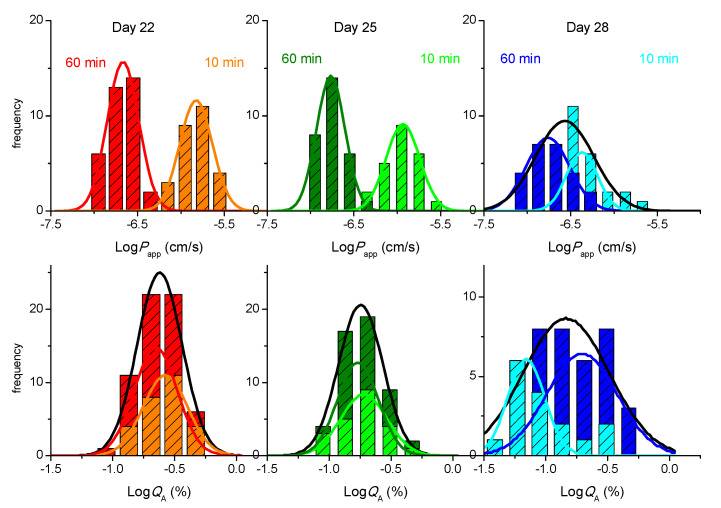
Effect of the sampling time interval on the paracellular permeability of LY through Caco-2 monolayers for cells at day 22, 25, or 28, post-seeding. The LY transport expressed as the Logarithm of the instantaneous permeability (Log*P*_app_) is shown in the upper plots, while that expressed as the Logarithm of the amount of LY that permeates (Log*Q*_A_) is shown in the lower plots, for a sampling interval of 10 min (light colors) or 60 min (dark colors). The lines are the best fit of a Normal distribution, with the parameters given in the Appendix A, the black line corresponding to the cumulative data from both sampling time intervals.

**Figure 5 pharmaceutics-13-01563-f005:**
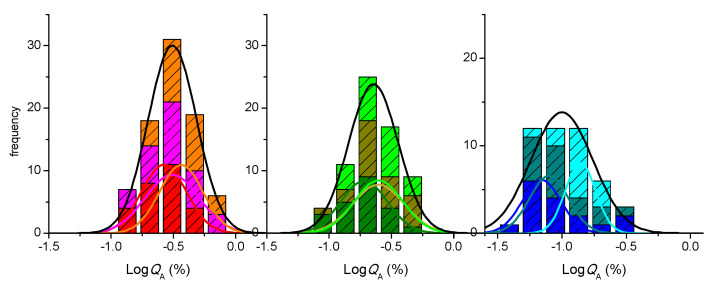
Effect of multi-time sampling on the paracellular permeability of LY through Caco-2 monolayers for cells at day 22, 25, or 28, post-seeding. The LY transport is expressed as the Logarithm of the amount of LY that reaches the acceptor compartment (Log*Q*_A_) for 3 consecutive 10 min sampling intervals (at 10, 20, and 30 min): 1st 10 min sampling (
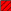
, 
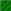
, 
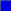
), 2nd 10 min sampling (
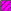
, 
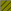
, 
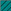
), and 3rd 10 min sampling (
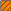
, 
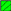
, 
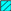
). The lines are the best fit of a Normal distribution, with the parameters given in Appendix A, the black line corresponding to the sum of the results from the 3 sampling time points.

**Figure 6 pharmaceutics-13-01563-f006:**
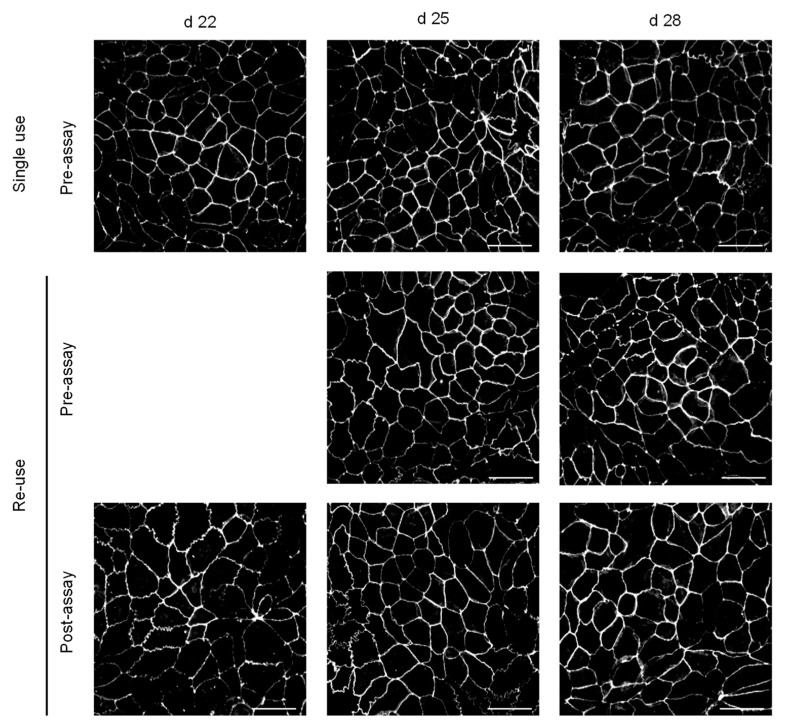
Immunofluorescence ZO-1 staining of Caco-2 monolayers. Representative images for cell monolayers at day 22, 25, and 28 are shown in the left, middle, and right panels, respectively. In the upper plots, the monolayer was not used in permeability assays, while in the additional plots the monolayers were previously used. The lower plots correspond to monolayers immediately after the LY permeability assay, and in the middle plots the images correspond to cell monolayers that were maintained in culture media for 2 days after the permeability assay. Scale bar 50 µm.

**Figure 7 pharmaceutics-13-01563-f007:**
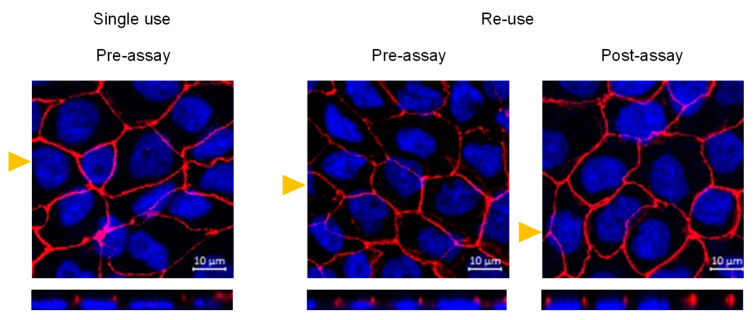
Staining of ZO-1 (red) and nuclei (blue) for Caco-2 monolayers at day 28 post-seeding. In the left plots, the monolayer was not used in permeability assays, in the middle plots the monolayer was previously used on days 22 and 25 and maintained in fresh media until day 28, and the right plots correspond to cell monolayer immediately after a permeability assay on day 28 (after being previously used on days 22 and 25). The lower plots correspond to the z-stacks at the cross section, indicated by the respective yellow triangle.

## Data Availability

The results obtained for all experiments performed are shown in the SI, the raw data will be provided upon request.

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
