# Peer review of "Re-Use of Caco-2 Monolayers in Permeability Assays—Validation Regarding Cell Monolayer Integrity"

_pharmaceutics, 2021, doi:10.3390/pharmaceutics13101563_

Round 1
Reviewer 1 Report
I have no suggesting. The work has good draw and it is easy reads
Author Response
We thank the reviewer for reading the manuscript and for the positive evaluation.
Reviewer 2 Report
Pires and coworkers revealed the full validation of the methodology proposed for the re-use of the Caco-2 cell monolayer from the results of the integrity and permeability assays at days 22, 25, and 28. The methodology proposed by the authors in this manuscript is fine-tuned compared to the method shown by Hubatsch et al. (Nat. Protoc. 2007). It will be better and faster protocols for Caco-2 users. It is acceptable that Caco-2 cells are broadly used and make a major contribution to pharmacological studies. However, there are problems in that some drug-metabolizing enzymes and drug transporters are absent or poorly expressed in Caco-2 cells [1,2]. Recently, a human intestinal organoid-derived monolayer is produced and used for a wide range of applications [3-5]. From the Caco-2 user, please give us opinions in discussion or conclusions.
- Sun, H.; Chow, E.C.; Liu, S.; Du, Y.; Pang, K.S. The Caco-2 cell monolayer: usefulness and limitation. Expert Opin. Drug Metab. Toxicol. 2008, 4, 395-411.
- Balimane, P.V.; Chong, S. Cell culture-based models for intestinal permeability: A critique. Drug Discov. Today 2005, 10, 335-343.
- Wang, Y.; DiSalvo, M.; Gunasekara, D.B.; Dutton, J.; Proctor, A.; Lebhar, M.S.; Williamson, I.A.; Speer, J.; Howard, R.L.; Smiddy, N.M.; et al. Self-renewing monolayer of primary colonic or rectal epithelial cells. Cell Mol. Gastroenterol. Hepatol. 2017, 4, 165-182.e7.
- Roodsant, T.; Navis, M.; Aknouch, I.; Renes, I.B.; van Elburg, R.M.; Pajkrt, D.; Wolthers, K.C.; Schultsz, C.; van der Ark, K.C.H.; Sridhar, A.; et al. A human 2D primary organoid-derived epithelial monolayer model to study host-pathogen interacton in the small intestine. Front. Cell. Infect. Mocrobiol. 2020, 10, 272.
5. Kozuka, K.; He, Y.; Koo-McCoy, S.; Kumaraswamy, P.; Nie, B.; Shaw, K.; Chan, P.; Leadbetter, K.; He, L.; Lewis, J.G.; et al. Development and characterization of a human and mouse intestinal epithelial cell monolayer platform. Stem Cell Reports 2017, 9, 1976-1990.
Author Response
please see file attached

Reviewer 3 Report
Dear Authors,
The manuscript entitled "Re-use of Caco-2 monolayers in permeability assays – validation regarding cell monolayer integrity" addresses the re-use of the Caco-2 cell cultures to increase the throughput. To validate the proposed methodology for re-use the Caco-2 cell cultures in permeability assays, the authors have evaluated the monolayer integrity and compared the results of the permeation assays obtained in the different days performed to those obtained from a single-use.
The manuscript is well-written, the introduction gives enough background on the topic and follows a logical flow, it also cites relevant references. The experimental work has been well-planned and well-executed, the methodology is well described and the results are presented clearly. I have a few comments:
- Figure 4: is there missing the black line in day 22 and day 25 in the upper plots?
- Figure B2: in my opinion, it is unclear what sampling time point corresponds to each. A detailed description should be given.
- Figure 5, the sampling time points 1st, 2nd and 3rd indicated in the legend is also unclear. Do they correspond to 10, 20 and 30 minutes?
- Line 540: The abbreviation CLSM is not described, it should appear for the first time in section 2.5.
- Line 600: please revise "where" in the sentence.
Kind regards,
Author Response
please see file attached

Reviewer 4 Report
The main scope of this study is to increase the throughput of the already well characterized and widely used Caco-2 intestinal model for drug screening. This manuscript tackles a relevant topic, as 3R become increasingly important in drug characterization and development, and provides an extensive protocol to extend the use of the Caco-2 model for drug permeability assays that is easy to follow. The successful application of the modified permeability model will certainly find further application in the future for high throughput drug screening. I would recommend accepting and publish the manuscript after small changes.
The in vitro model will be mainly used to study permeability of mainly uncharacterized substances. Before the in vitro model can be used for screening poorly defined substances, it remains to be investigated whether the increasing integrity of the cell monolayer between days 22 and 28 and any previous incubation with other substances alters the final determined permeability of the drugs. The authors of the study have already begun to characterize this with lucifer yellow. Nevertheless, it would have been nice to also apply the re-used model to already tested substances in nontoxic concentrations to test whether tested permeability of the cell monolayer changes when used several times or or is comparable to previous publications.
Moreover, it would have been nice to already compare the permeability and the determined Papp of other already investigated and well characterized substances, for which the transport path (apical to basolateral and basolateral to apical) plays an essential role (also for the ultimately physiologically remaining concentration of the substance in the body), such as glycyl-sarcosine, atropine, or digoxin (Hubatsch, I., et al. (2007). "Determination of drug permeability and prediction of drug absorption in Caco-2 monolayers." Nature Protocols 2(9): 2111-2119.). I understand if this is beyond the scope of the current publication, as it is already very extensive, but would definitely recommend to include this in later studies.
Lines 254-257 and Figure 6: In the main text the authors mention that fluorescence images were also captured in Z-stacks. I would recommend also adding the optical cross-sections (x–z direction) with ZO-1 staining, similar to the Nature protocol from Artursson and collaborators. I believe that the inclusion of Z-stack images of ZO-1 in monolayers of Caco-2 cells would further highlight the result of this experiment by providing additional insight into the intracellular distribution of the protein. ZO-1 is peripheral tight junction protein and predominantly expressed to the apical side of the cells, yet to clearly exclude changes/redistribution of ZO-1 in reused or/and "older" monolayers, the apical or lateral localization of ZO-1 in Z-stacks should be shown additionally. Furthermore, the representation of Z-stacks would also give the possibility to show the images as co-staining of ZO-1 together with the staining of the nucleus.
Author Response
please see file attached
